# Identification of 2,4-Diaminoquinazoline Derivative as a Potential Small-Molecule Inhibitor against Chikungunya and Ross River Viruses

**DOI:** 10.3390/v15112194

**Published:** 2023-10-31

**Authors:** Amrita Saha, Badri Narayan Acharya, Manmohan Parida, Nandita Saxena, Jaya Rajaiya, Paban Kumar Dash

**Affiliations:** 1Virology Division, Defence Research & Development Establishment, Gwalior 474002, India; avadher@salud.unm.edu (A.S.); paridamm.drde@gov.in (M.P.); 2Department of Molecular Genetics and Microbiology, University of New Mexico Health Sciences Center, Albuquerque, NM 87131, USA; jrajaiya@salud.unm.edu; 3Synthetic Chemistry Division, Defence Research & Development Establishment, Gwalior 474002, India; bnacharya.drde@gov.in; 4Pharmacology & Toxicology Division, Defence Research & Development Establishment, Gwalior 474002, India; nanditasaxena.drde@gov.in

**Keywords:** chikungunya virus, antiviral drug, 2,4-diaminoquinazoline derivative, Ross River virus, pan-alphavirus inhibitor, drug screening

## Abstract

Alphaviruses are serious zoonotic threats responsible for significant morbidity, causing arthritis or encephalitis. So far, no licensed drugs or vaccines are available to combat alphaviral infections. About 300,000 chikungunya virus (CHIKV) infections have been reported in 2023, with more than 300 deaths, including reports of a few cases in the USA as well. The discovery and development of small-molecule drugs have been revolutionized over the last decade. Here, we employed a cell-based screening approach using a series of in-house small-molecule libraries to test for their ability to inhibit CHIKV replication. DCR 137, a quinazoline derivative, was found to be the most potent inhibitor of CHIKV replication in our screening assay. Both, the cytopathic effect, and immunofluorescence of infected cells were reduced in a dose-dependent manner with DCR 137 post-treatment. Most importantly, DCR 137 was more protective than the traditional ribavirin drug and reduced CHIKV plaque-forming units by several log units. CHIKV-E2 protein levels were also reduced in a dose-dependent manner. Further, DCR 137 was probed for its antiviral activity against another alphavirus, the Ross River virus, which revealed effective inhibition of viral replication. These results led to the identification of a potential quinazoline candidate for future optimization that might act as a pan-alphavirus inhibitor.

## 1. Introduction

The alphaviruses occur worldwide and include approximately 30 different species of positive-sense single-stranded RNA viruses causing significant diseases such as arthritis or encephalitis in animals and humans [1,2,3]. Important members of arthritogenic alphaviruses include chikungunya virus (CHIKV), Ross River virus (RRV), Sindbis virus (SINV), and Semliki Forest virus (SFV), while encephalitic alphaviruses include Venezuelan equine encephalitis (VEEV), Eastern equine encephalitis virus (EEEV), and Western equine encephalitis virus (WEEV). Several alphaviruses are significant human health threats, including CHIKV, RRV, and SINV, while VEEV, WEEV, and EEEV are zoonotic threats and designated as category B agents by the CDC, which is indicative of the level of serious threat [3,4]. 

CHIKV has been responsible for large-scale human morbidity across the continents over the last decade, causing significant, long-lasting polyarthralgia [5,6,7,8]. Due to inadequacies in vector control measures and failure in the development of an approved vaccine, the necessity to develop efficacious, antiviral therapeutics is urgently needed. The unmet clinical need for chikungunya fever is the underlying driving motivation for this study. At present, only two antiviral compounds (chloroquine and ribavirin) that inhibit CHIKV infection in vitro have been explored in clinical studies. However, neither of these compounds has shown strong efficacy in clinical trials [9,10,11,12]. Therefore, researchers are exploring newer approaches that have been proven efficacious for other viral diseases.

The discovery and development of small-molecule (SM) drugs have been used in other viral diseases [13,14,15,16,17,18,19,20,21]. Small-molecule inhibitors (SMIs) are low-molecular-weight compounds ranging up to 500 Da. Their small size allows for easy penetration to reach the target within the cell for inhibition. SMs are of particular interest because of their ease of accessibility to a variety of desired targets, their convenient handling due to their reasonable stability, and, most importantly, because they are best suited for oral administration. SMIs have already been used for influenza treatment successfully [19,20,21]. So far, many SM drugs have been approved, such as oseltamivir, zanamivir, peramivir, amantadine, and rimantadine, along with several other drugs in various stages of development [22]. 

The present study aimed to identify anti-CHIKV compounds, for which a cell-based screening approach was employed. The virus-induced cytopathic effect (CPE) of the cell and the cell viability assay as a readout were used to assess the efficacy of various compounds used in the in vitro screening assay. The in-house SM library, which has nine different families of compounds (viz., pyridine-quinolone hybrids, hydrazones of nicotic acid and isonicotinic acid, triaryl pyrazoline, oxazin-3-one, oxadiazoles, dihydropyrimidine dione, xanthenes, quinazoline derivatives, and substituted benzonitriles), was probed for its anti-CHIKV activity. A quinazoline derivative (6-fluoro-quinazoline-2,4-diamine; DCR 137) was identified through in vitro screening and demonstrated greater protection compared to that of the existing drug, ribavirin, against CHIKV. The absence of broad-spectrum antivirals is a major challenge affecting the treatment of viral infections. The successful development of broad-spectrum molecules having activity against genera or families of viruses is of paramount importance. Keeping this in view, this molecule was further evaluated for its broad-spectrum activity by studying the inhibition of another alphavirus, RRV, which is emerging in many parts of Oceania. 

## 2. Material and Methods

### 2.1. Cells and Viruses

Vero cells (African green monkey kidney cell line) were obtained from the National Center for Cell Science (NCCS), Pune, India, and were cultured in minimum essential medium (MEM) supplemented with 10% heat-inactivated fetal bovine serum (FBS), 80 U gentamicin, 2 mM L-glutamine, and 1.1 g/L sodium bicarbonate. CHIKV DRDE-07 strain (GenBank accession no. EU372006) and RRV T48 strain (GenBank accession no. GQ433359) were propagated using standard virus adsorption techniques, titrated by plaque assay, and stored at −80 °C as previously described [23,24].

### 2.2. DRDE Chemical Repository (DCR) Library

A collection of 150 small molecules belonging to different classes of medically important drug-like chemical compounds was sourced from the DCR library (Table 1). The reference compound ribavirin (RIBA) was purchased from Sigma, St. Louis, MO, USA.

### 2.3. Preparation of Compound Stock Solution 

For cytotoxicity studies, each compound of the DCR series was weighed separately, dissolved in 100 µL DMSO, and volume was made up to 1 mL with MEM to obtain a stock solution of concentration 100 mM. The stock solution was sterilized by filtration, and a serial two-fold dilution was prepared to obtain lower concentrations.

### 2.4. Cytotoxicity Assay for Screening Small Molecules

Vero cells (1 × 10^4^ cells per well of a 96-well plate) were treated with different concentrations of two-fold diluted compounds for 72 h. The cytotoxic effects of all the compounds were determined based on an MTT (3-(4,5-dimethylthiazol 2-yl)-2,5-diphenyltetrazolium bromide) assay. The MTT reagent (0.5 mg/mL) was added 72 h post-treatment of the compounds. Finally, DMSO was added, and the absorbance was measured at 570 nm. All the readings were normalized with the control experiment, in which compounds were not added. The maximum nontoxic doses (MNTD) of all the compounds were calculated. The percentage of cell viability was calculated as follows: 100% − (absorbance of treated cells/absorbance of untreated cells) × 100%. The concentration required to reduce 50% cell viability (CC_50_) was determined.

### 2.5. In Vitro Screening of Compounds against CHIKV

A cell-based virus inhibition assay using the Vero cell line was carried out for the preliminary screening of the compounds for their anti-CHIKV activity based on cytopathic effect (CPE) inhibition and cell viability assays. Briefly, Vero cells were seeded in 96-well plates and incubated at 37 °C with 5% CO_2_ in a humidified incubator for 24 h. Confluent Vero cells were infected with MOI 0.01 of CHIKV DRDE-07, and following 2 h of adsorption, cells were incubated with different concentrations of two-fold diluted compounds (DCR series) for 72 h. Ribavirin was used as the positive control. The CPE was observed for 24–72 hpi by analyzing cell and nuclear morphology under an inverted light microscope (Leica, Mannheim, Germany). After 72 h, cell viability was assessed by MTT assay. The best-shortlisted compound, DCR 137, was further synthesized with a good yield, as described previously [25]. It was characterized by mass spectrometry and ^1^H-nuclear magnetic resonance spectroscopy using the Micromass Q-ToF mass spectrometer and the Bruker AV 400 NMR spectrometer, respectively.

### 2.6. Evaluation of Antiviral Activity of DCR-137 against CHIKV

#### 2.6.1. Mode of Inhibition

Vero cells were seeded at a cell density of 1 × 10^4^ cells per well of a 96-well plate and incubated overnight. Cells were infected with CHIKV at an MOI of 1. Compound (300 µM DCR 137) was added 24 h before infection (in pre-treatment), 0 h after infection (in simultaneous treatment), and 2 h after viral adsorption (in post-treatment). For the pre-treatment group, the compound was added to the cells 24 h prior to infection, and the compound was removed by washing the cells before infection; for the simultaneous-treatment group, the compound and CHIKV were administered at the same time; for the post-treatment group, the virus was added to the cells, and after 1 h of incubation for viral attachment, the inoculum was removed. Finally, the medium containing compound was added 2 h after the infection of cells with CHIKV post-treatment. In the post-treatment mode, the drug was maintained in culture until the supernatants and cells were harvested. The plates were incubated at 37 °C with 5% CO_2_ for 48 h, and CPE was monitored at different time intervals from 12 h to 48 h. Cell viability was measured by MTT assay at 48 hpi. 

#### 2.6.2. Cytopathic Effect Inhibition and Cell Viability Assay

Vero cells were seeded onto 24-well culture plates at a cell density of 1 × 10^5^ cells per well and allowed for overnight incubation for cell attachment. Cells were infected with CHIKV at an MOI of 0.1 in a post-treatment mode. Serial two-fold dilution of the compound was prepared in MEM and added in triplicate. The plates were incubated at 37 °C, 5% CO_2_ for 48 h. Microscopic examination of plates for CPE reduction was monitored at different time intervals. The cytotoxic effect of the compound was further determined by a cell viability assay (MTT assay, Sigma, USA). After 48 h of incubation, MTT solution (0.5 mg/mL) was added to the cultures and incubated for an additional 3 h at 37 °C. Finally, dimethyl sulfoxide (DMSO) was added, and the absorbance intensity was measured by a microtiter plate reader (Synergy H1 BioTek, Winooski, VT, USA) at 570 nm. Cell control was treated only with medium without the compound, and virus control was kept alongside. Virus-infected cells treated with ribavirin were used as a positive control. The dose–response assay was designed to determine the range of efficacy of DCR 137 against CHIKV. The half-maximal effective concentration (EC_50_) for compounds was calculated. A selectivity index (SI) value was calculated, which is CC_50_/EC_50_. A compound with an SI value ≥ 10 is generally considered to be active in vitro.

#### 2.6.3. Plaque Assay

Cells were seeded in a 25 cm^2^ cell culture flask, and on confluency, cells were infected with CHIKV at an MOI of 0.1. DCR 137 was added at a 300 µM concentration after 2 h of virus adsorption. Infection was allowed to proceed for 48 h. The culture supernatant was collected at 24 hpi and 48 hpi, and virus yield was determined by the plaque assay as described earlier [23]. The effectiveness of the drug in reducing the plaque-forming units (PFUs) of virus compared with controls was calculated considering the volume and dilution factor of the inoculum.

#### 2.6.4. Immunofluorescence Assay

The effect of compound treatment on CHIKV was assessed by immunofluorescence assay using CHIKV E2 mAb [26], as described earlier [23]. Briefly, Vero cells were seeded on coverslips in a 6-well plate. Compound treatment (300 µM) and virus infection (MOI 0.1) methods were the same as described above. Slides were examined, and images were captured with a fluorescence microscope (Leica, Mannheim, Germany) equipped with FITC and DAPI-compatible filters.

#### 2.6.5. Flow Cytometric Analysis

Cells were infected with CHIKV at an MOI of 0.1 in a 75 cm^2^ cell culture flask, followed by treatment with 300 µM DCR 137. Infection was allowed to proceed for 24 hpi. Flow cytometric analysis was performed as previously described [27].

#### 2.6.6. Western Blot

The mock Vero cells and CHIKV-infected cells (MOI 0.1) with and without drug treatment were harvested at 18 hpi. Cells were lysed using a RIPA buffer/protease inhibitor cocktail. After quantitation, 50 µg of protein was run in 10% SDS-PAGE, electroblotted onto PVDF membrane, and analyzed using CHIKV E2 mAb as described previously [23].

### 2.7. Evaluation of Antiviral Activity of DCR-137 against RRV

Next, we examined whether compound DCR 137 could inhibit other alphaviruses or not. The efficacy of DCR 137 against RRV was tested at an MOI of 0.1. The effect of DCR 137 on the inhibition of RRV infection in Vero cells was assessed by reduction in CPE, cell viability assay, plaque reduction, and immunofluorescence assay (using mouse anti-native RRV polyclonal antibodies) as described above for the CHIKV study.

### 2.8. Statistical Analyses

All the assays were performed three times with each sample in triplicate, and the results were graphed, with error bars indicating the mean ± standard deviation (SD). Data were analyzed using Student’s *t*-test. The asterisk indicates statistical significance (* *p*-value < 0.05, ** *p*-value < 0.01, *** *p*-value < 0.001).

## 3. Results

### 3.1. In Vitro Antiviral Screening of Small Molecules

Prior to evaluating the anti-CHIKV properties of the DCR library, compounds were subjected to toxicity studies in order to determine the maximal dose, which could be non-toxic to the cells. The studies were initiated by using two-fold serially diluted compounds to achieve a specific cytotoxic concentration. Compound concentrations with more than 90% cell viability were chosen as the highest concentration (MNTD_90_). The MNTD of each compound obtained in Vero cells is presented in Appendix A. A high MNTD value indicates that the compound is less toxic as compared to the others with low MNTD values. The in vitro antiviral assay was initiated using the MNTD of each compound against CHIKV. In vitro antiviral screening identified nine SMIs that inhibited CHIKV replication along with a >50% cell survival rate. These compounds belonged to two different series: Pyridine-quinolone hybrid and quinazoline. Synthesis of these two series of compounds has been described previously [25,28]. On testing all nine compounds, DCR 137 inhibited CHIKV replication while preserving more than 90% cell survival at 72 hpi (Table 2; Figure 1A). The mass spectrum of DCR 137 was recorded with an M + 1 mass peak of m/z 178.9613 (Figure 1B). The purity of the compound was further tested by ^1^H-NMR and showed characteristic spectral data (Figure 1C). Finally, this compound was further evaluated, and its antiviral activity against CHIKV was validated with six different techniques, viz., reduction in CRE, cell viability assay (MTT assay), immunofluorescence assay, plaque assay, Western blot, and flow cytometric analysis.

### 3.2. Evaluation of Antiviral Activity of DCR-137 against CHIKV

The cytotoxic effect of DCR 137 was first tested in Vero cells. The MNTD value determined in this study was >300 μM (~312 μM), which showed ~90% cell viability (Figure 2). Therefore, all subsequent in vitro experiments were carried out using a concentration less than or equal to 300 μM. The CC_50_ concentration of DCR 137 in Vero cells was found to be >1100 µM. Mode of inhibition studies revealed that post-treatment with DCR 137 was most effective by preserving more than 95% cell viability as compared to pre-treatment (15%) and simultaneous treatment (29%) (Figure 3A). Antiviral activity in the presence of various concentrations of DCR 137 in a post-treatment mode at 48 hpi was assessed by performing an MTT assay on CHIKV-infected Vero cells and showed inhibition in a dose-dependent manner (Figure 3B). Based on the absorbance at 570 nm of DCR 137-treated and untreated cells post-infection, the EC50 and selectivity index (SI) were calculated to be ~37 µM and 31, respectively. Microscopic examination of CHIKV-infected Vero cells revealed a marked cytopathic effect (CPE), whereas cells treated with 150 and 300 µM DCR 137 showed delayed and dose-dependent CPE, with a higher concentration of DCR 137 showing lesser CPE as compared to virus control (Figure 4). The inhibitory effect of compounds was also confirmed by the detection of intracellular viral load by immunofluorescence assay. The DAPI staining showed viable cells in all groups. DCR 137-treated infected cells exhibited lower levels of fluorescence intensity, with a complete abrogation of viral fluorescence in the DCR 137-treated group at 300 µM. Intriguingly, DCR 137 was found to have higher protection when compared to ribavirin, which was used as a positive control (Figure 4B). Next, the antiviral activity of DCR 137 against CHIKV was investigated by plaque assay. Results showed that the CHIKV titer expressed as plaque-forming units (PFU) per mL was significantly reduced following treatment with 300 μM of DCR 137 compared with untreated cells. CHIKV titers in DCR 137-treated cells were 1.8 × 10^2^ PFU/mL and 1.9 × 10^5^ PFU/mL, compared to 2.8 × 10^6^ and 9.1 × 10^8^ PFU/mL in virus control at 24 hpi and 48 hpi, respectively. Percent inhibition with respect to virus control was calculated, and the results showed that the viral titer was reduced nearly to 99.99% and 99.97% at 24 hpi and 48 hpi, respectively, in cells post-treatment with 300 μM DCR 137 (Figure 5A). Western blot analysis also revealed a significant decrease in CHIKV-E2 protein (~48 kDa) expression in a dose-dependent manner compared to virus control after treatment with DCR 137. The E2 protein band was not observed in mock cell control, whereas as a load control, β-actin protein expression was equivalent in all groups (Figure 5B). The cell population as assessed by flow cytometry represents the mean fluorescence-positive cells (CHIKV-E2/FITC expression). The flow cytometric analysis showed a decrease in mean fluorescence intensity in DCR 137-treated cells compared to virus control, indicating a significant decrease (*** *p*-value < 0.001) in viral replication after DCR 137 treatment (Figure 5C).

### 3.3. Evaluation of Antiviral Activity of DCR-137 against RRV

We further extended this study to another alphavirus, RRV replication in Vero cells, and evaluated the efficacy of the inhibitor DCR 137. Vero cells were post-treated with DCR 137 after infection with the RRV T48 strain at an MOI of 0.1 (Figure 6). The virus-induced CPE was recorded at 48 hpi by a light microscope (Figure 6A). Further, indirect immunofluorescence assay results indicated a low expression of viral proteins in the treated group (300 µM and 150 µM DCR 137) compared to the control in a dose-dependent manner, as assessed with RRV-FITC polyclonal antibodies (Figure 6B). This indicates that DCR 137 treatment suppressed RRV infection significantly compared with virus control and the traditional drug ribavirin used as a positive control in the study. Notably, the green fluorescence was barely detectable in the cells treated with 300 µM DCR 137 at 36 hpi (Figure 6B). The MTT assay of RRV-infected Vero cells revealed reduced cell viability in the presence of DCR 137 (Figure 6C). In addition, the robust inhibitory potential of DCR 137 was also confirmed by a plaque assay, which demonstrated a reduction in viral titer in comparison to mock-infected Vero cells when treated with DCR 137 up to 48 h. The viral titers in DCR 137-treated cells at 24 hpi and 48 hpi were 1.1 × 10^2^ PFU/mL and 1.7 × 10^5^ PFU/mL, respectively, in contrast to the titers of 3.0 × 10^6^ and 1.2 × 10^8^ PFU/mL in untreated cells at 24 hpi and 48 hpi, respectively. Percent inhibition calculated with respect to virus control showed a 99.99% and 99.85% reduction in virus titer by 300 µM DCR 137 at 24 hpi and 48 hpi, respectively (Figure 6D).

## 4. Discussion

Alphavirus infections are neglected tropical diseases for which medical countermeasures are not available. Alphaviruses are medically important arboviruses that can cause fevers, rash, and rheumatic diseases (CHIKV, ONNV, RRV) or potentially fatal encephalitis (EEEV, VEEV, WEEV) [29,30]. Arthritogenic alphaviruses such as chikungunya virus (CHIKV) and Ross River virus (RRV) are responsible for large-scale epidemics. These alphaviruses are emerging pathogens that have been progressively expanding their global distribution. CHIKV has recently emerged to cause millions of human infections worldwide. The pandemic potential of CHIKV has long been recognized; in 2018, CHIKV was added to the WHO shortlist for priority research and development, which notably also included pandemic coronaviruses [31]. RRV has recently been suggested to be a potential emerging infectious disease worldwide. RRV infection remains the most common human arboviral disease in Australia, with a yearly estimated economic cost of USD 4.3 billion [32]. The primary goal of this study was to identify small-molecule inhibitors against CHIKV, which may later be extended to identify broad-spectrum antivirals.

One of the initial and most crucial steps of a drug development program involves the pre-clinical screening of a large number of compounds, preferably small-molecule libraries, to develop a potential therapeutic countermeasure against an infection. CHIKV is capable of replication in vitro in various mammalian cells with significant cytopathic effects. Therefore, to identify anti-CHIKV compounds, we have employed a cell-based screening approach to probe various small molecules for their ability to inhibit CHIKV by measuring the reduction in CPE, cell viability, viral protein expression, and plaque assay as a readout. One of the quinazoline derivative compounds, named DCR 137, was found to be the most potent against CHIKV.

The antiviral activity of DCR 137 was examined in Vero cells. Reduction in CPE and immunofluorescence assay revealed higher virus inhibition potential by DCR 137 compared to the reference compound ribavirin. DCR 137 was found to inhibit CHIKV replication significantly by 99.99% and 99.97% at 24 hpi and 48 hpi, respectively, as assessed by the plaque reduction assay. Then, a time-of-addition assay was performed to identify which step of the viral life cycle is blocked by DCR 137. Vero cells were infected with CHIKV at an MOI of 1, and the compound was added before and after infection. Pre-treatment with DCR 137 for 24 h prior to CHIKV or simultaneous infection with the virus for 1 h showed no inhibitory effects on viral infection, indicating that DCR 137 did not inhibit virus attachment, entry, or disassembly processes. A maximal reduction in viral titers was observed when DCR 137 was added at 2 h post-treatment. These results indicate that DCR 137 may exert its effect during viral replication/maturation (post-entry step of the CHIKV life cycle). The specific mechanism by which this compound affects the post-entry step of the CHIKV life cycle effectively is not known, but there is a possibility that DCR 137 might inhibit a host factor crucial for viral replication.

There is currently a lack of both narrow- and broad-spectrum antivirals available to combat arboviral pathogens. Broad-spectrum antivirals are attractive therapeutics, offering the promise of a single antiviral to combat several viral pathogens [33], and could be effective in controlling emerging and novel pathogens. Further, they are crucial when a quick and specific diagnosis is not available. As such, the problem of overlapping clinical manifestations makes the clinical diagnosis of arboviral infections extremely difficult. For this, the compound studied here, DCR 137, a diamino quinazoline derivative, was further probed for its broad-spectrum activity against another alphavirus, RRV, and the results revealed a percent inhibition of 99.99. These results strongly suggest that DCR 137 is not only a potential candidate to inhibit CHIKV but can also act as a broad-spectrum antiviral as a pan-alphavirus inhibitor. RRV is an old-world alphavirus that has been grouped with CHIKV [5]. They both share very similar clinical manifestations and are primarily arthritogenic [34,35]. CHIKV and RRV, along with some other alphaviruses, also share a common entry mediator, Mxra8 [36]. Although knowledge of the life cycle and pathogenesis of both RRV and CHIKV infections is advancing rapidly, many similarities and differences between them still need to be explored.

Quinazolines are a large class of biologically active compounds that exhibit a broad spectrum of biological activities against viruses, bacteria, and fungi, as well as anticancer, anti-inflammatory, antimalarial, and antioxidant effects [37,38,39]. FDA-approved quinazolines constitute a promising group of drugs against different types of tumors in cancer and mechanistically inhibit the protein kinase of the epidermal growth factor receptor (EGFR), inactivating the anti-apoptotic pathway [40,41,42,43]. Thus, in our study, there is a strong possibility that DCR 137 inhibits post-viral replication via its effect on specific host factor(s) important to viral replication. Also, a drug that effectively targets a common host factor is likely to be effective against different viruses and decrease the risk of developing drug resistance [33].

Moreover, during the review process, we became aware that Chao and co-workers addressed a similar type of 2,4-diaminoquinazoline derivative that showed potent activity against Dengue virus (DENV) with an excellent pharmacokinetic profile [44]. Chao et al. reported this compound to be the most potent inhibitor uncovered to date using a cell-based DENV replicon assay. This work, together with the previous report [44], strengthens the belief that a single quinazoline-based drug molecule can be effective against two medically important families of arboviruses, viz., alphaviruses and flaviviruses. The possible improvements in activity can be further achieved by minor modifications in the basic quinazoline nucleus to obtain derivatives that may show better effects and less toxicity. Our current lead compound, DCR 137, has a low molecular weight (~178 Da) and is amenable to further “hit-to-lead” optimization to achieve appropriate drug metabolism and pharmacokinetic properties. Thus, our finding that DCR 137 is a very promising antiviral can serve as a design to produce an improved broad-spectrum antiviral in the future.

## 5. Limitations of the Study and Future Directions

While the compound DCR 137 identified in this study holds promising potential to be developed as an inhibitor of CHIKV and RRV infections, there are a few limitations that need to be addressed before moving DCR 137 or its structural analog(s) forward to the clinic. First, animal studies are needed to test the in vivo toxicity, efficacy, and pharmacokinetics of DCR 137 or its structural analogs. Moreover, it would be helpful to test the efficacy of DCR 137 in inhibiting the viral infection of various other alphaviruses. To broaden the application potential of DCR 137, it would be interesting to test if DCR 137 can work synergistically with other therapeutics. Mechanistically, this study suggests that compound DCR 137 inhibits viral replication/maturation. However, a more detailed mechanistic study is needed to understand the exact mode of action of the compound. It would be important to identify viral and/or host factor(s) binding DCR 137 in the future. Thus, further mechanism of action studies and in vivo experiments are needed to confirm the systemic effect of compound DCR 137 in the alphavirus life cycle.

## Figures and Tables

**Figure 1 viruses-15-02194-f001:**
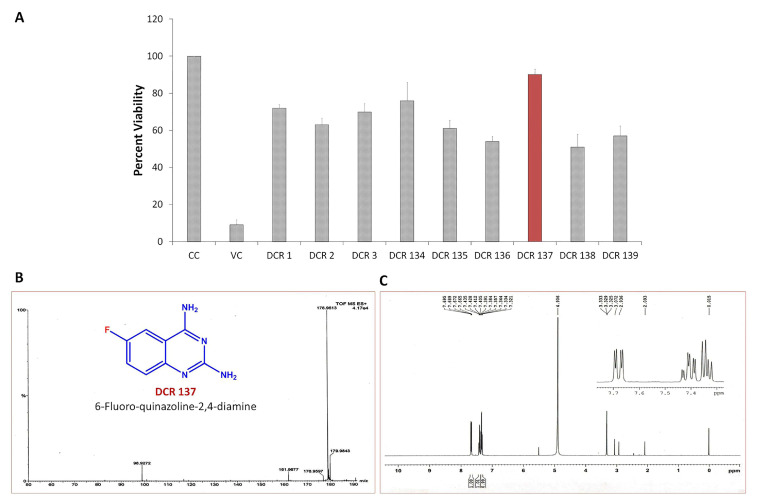
(**A**) In vitro screening of anti-CHIKV compounds (using MNTD as indicated in Table 2) by measurement of cell viability as readout gave rise to nine SMs that inhibited CHIKV replication by preserving >50% cell survival rate at 72 hpi. (**B**) MS and (**C**) ^1^H-NMR spectra of best shortlisted compound DCR 137.

**Figure 2 viruses-15-02194-f002:**
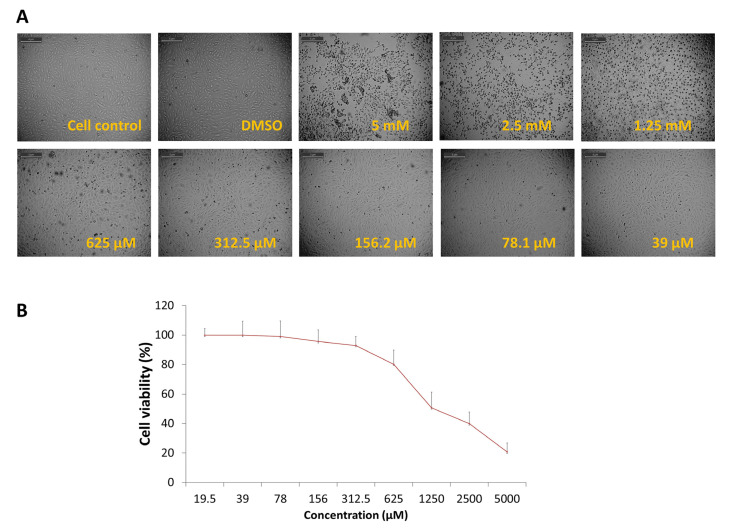
Cytotoxicity assay and determination of MNTD of DCR 137 in Vero cells. Cells were treated with different concentrations of DCR 137. (**A**) Microscopic images showing morphology of treated Vero cells at 48 hpi. No significant cytotoxicity was observed at 312.5 µM DCR 137. Data shown here are representative of one of the three experimental repeats. (**B**) Maximum non-toxic dose (MNTD) of DCR 137 was calculated through cell viability MTT assay on Vero cells.

**Figure 3 viruses-15-02194-f003:**
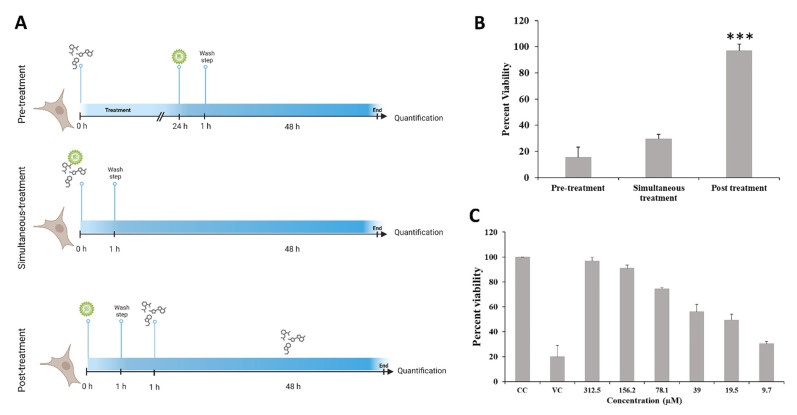
Mode of inhibition of DCR 137. (**A**) Schematic illustration of the time-of-addition/mode of inhibition experiment. Vero cells were infected with CHIKV at an MOI of 1, and treated with DCR 137 pre (−24 h), during (0–1 h) and post (2 h) infection. DMSO at 0.1% was added at the same time as a control. (**B**) Post-treatment mode showed significant viral inhibition after treatment with 300 µM DCR 137 compared to pre-treatment and simultaneous treatment. (**C**) Inhibition of CHIKV post-treatment with DCR 137 in a dose-dependent manner was assessed by MTT assay at 48 hpi to determine the percent viability after treatment. Data represent the mean ± SD of three independent experiments. The asterisk indicates statistical significance (*** *p* < 0.001).

**Figure 4 viruses-15-02194-f004:**
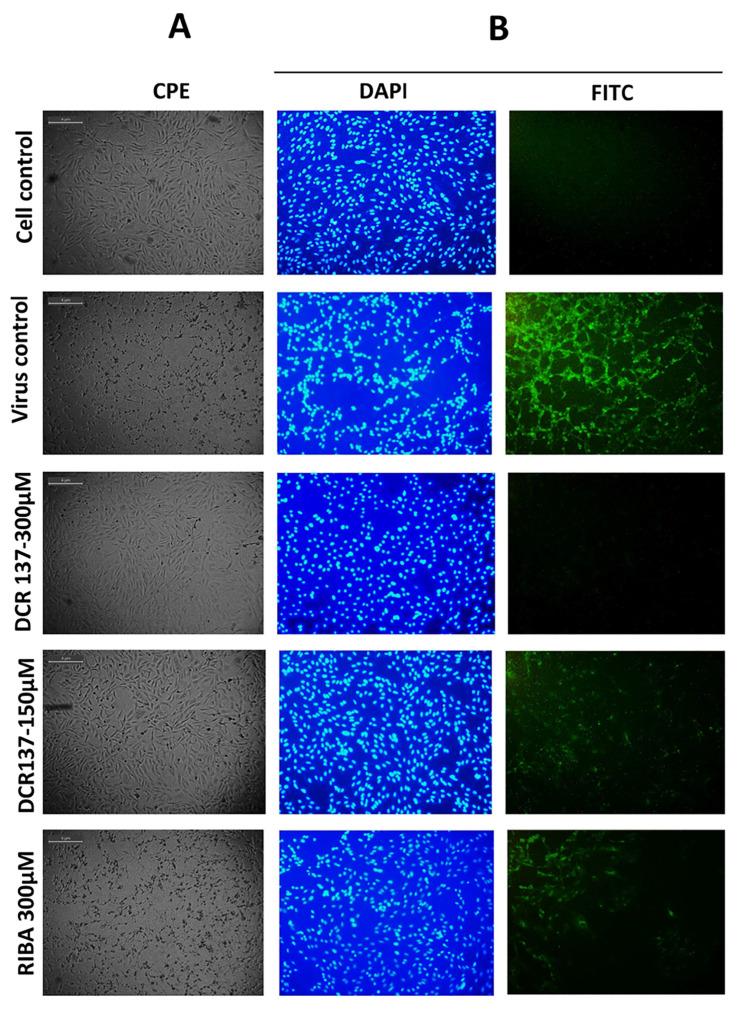
Reduction in CPE and immunofluorescence assay. Vero cells were infected with CHIKV, followed by addition of different concentrations of DCR 137 post 2 h virus adsorption. (**A**) Microscopic images showing cytopathic effect at 48 hpi after respective treatment. (**B**) Immunofluorescence assay: Cells were observed at 36 hpi, green fluorescence indicates the virus load as assessed with anti-CHIKV E2 mAb and secondary antibody conjugated with FITC, whereas blue fluorescence indicates the nuclear staining with DAPI. Data shown here are representative of one of the three experimental repeats.

**Figure 5 viruses-15-02194-f005:**
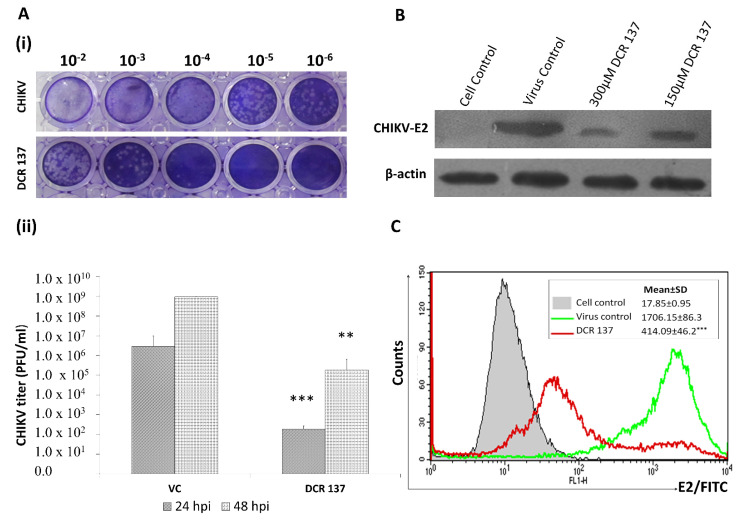
Plaque reduction assay, Western blot, and flow cytometry assay. (**A**) (i) Representative image of the plaques for the CHIKV load in cell supernatant at 48 hpi. (ii) CHIKV-infected cells were treated with 300 µM of DCR 137. Cell supernatant was collected at 24 hpi and 48 hpi. Virus titer was assessed by plaque assay. Plaques were visualized by staining with crystal violet. Titer was calculated considering the volume and dilution factor of the inoculum. Data represent the mean ± SD of three independent experiments. The asterisk indicates statistical significance (** *p* < 0.01, *** *p* < 0.001). (**B**) CHIKV E2 expression was analyzed by Western blot, which revealed significant inhibition of CHIKV in a dose-dependent manner after treatment. β-actin served as a loading control. (**C**) Analysis of reduction in CHIKV infectivity level after treatment with 300 µM DCR 137 by flow cytometry; filled histogram is control sample (control cell + DMSO). The mean fluorescence intensity (MFI) of CHIKV E2 expression in each group was plotted. Mean ± SD of MFI (*** *p* < 0.001) of three independent experiments is shown.

**Figure 6 viruses-15-02194-f006:**
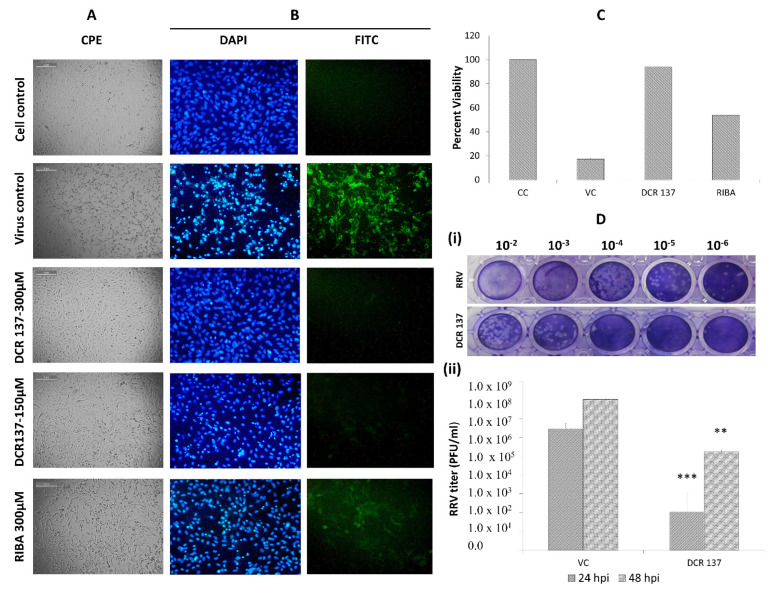
Antiviral activity of DCR137 against RRV. (**A**) Microscopic images showing cytopathic effect at 48 hpi after respective treatment. (**B**) Immunofluorescence assay: Cells were observed at 36 hpi, green fluorescence indicates the virus load as assessed with mouse anti-native RRV polyclonal antibodies and secondary antibodies conjugated with FITC; and blue fluorescence indicates the nuclear staining with DAPI at 20X. Data shown here are representative of one of the three experimental repeats. (**C**) Vero cells were infected with RRV, followed by addition of 300 µM DCR 137 post-2 h virus adsorption, MTT assay was performed 48 hpi to determine the percent viability after treatment. (**D**) (i) Representative image of the plaques for the RRV load in cell supernatant at 48 hpi. (ii) RRV-infected cells were treated with 300 µM of DCR 137. Cell supernatant was collected at 24 hpi and 48 hpi. Virus titer was assessed by plaque assay. Data represent the mean ± SD of three independent experiments. The asterisk indicates statistical significance (** *p* < 0.01, *** *p* < 0.001).

**Table 1 viruses-15-02194-t001:** DCR library with different series of small molecule compounds.

S. No.	DCR Code	Series Type
1.	DCR 1-DCR 3	Pyridine-quinolone hybrids
2.	DCR 4-DCR 53	Hydrazones of nicotic acid and isonicotinic acid
3.	DCR 54-DCR 63	Triaryl pyrazoline
4.	DCR 64-DCR 71	Oxazin-3-one
5.	DCR 72-DCR 85	Oxadiazoles
6.	DCR 86-DCR 112	Dihydropyrimidine dione
7.	DCR 113-DCR 133	Xanthenes
8.	DCR 134-DCR 139	Quinazolines
9.	DCR 140-DCR 150	Substituted benzonitriles

**Table 2 viruses-15-02194-t002:** List of best-shortlisted compounds that inhibit the replication of CHIKV.

DCR Code	Compound	Systematic Name	Structure	Group	Mol.Wt. (Da)	MNTD	CC_50_, EC_50_, SI	Cell Viability **
DCR 1	C_14_H_7_BrN_2_O_2_	5-bromo-7,12-dioxa-1,11-diaza-benzo[a]anthracene	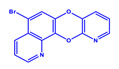	Pyridine-quinolone hybrids	315.12	50 µM	n.d. *	72%
DCR 2	C_18_H_8_Br_2_N_2_O_2_	5,12-dibromo-[1,4]dioxino[2,3-h:5,6-h’]diquinoline	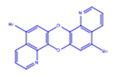	Pyridine-quinolone hybrids	444.07	50 µM	n.d.	63%
DCR 3	C_19_H_10_ClN_3_O_3_	7-[(2-chloropridin-3-yl)oxy]-5,12-dioxa-1,11-diazatetraphene	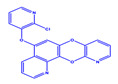	Pyridine-quinolone hybrids	363.754	50 µM	n.d.	69.9%
DCR 134	C_8_H_8_N_4_	Quinazoline-2,4-diamine	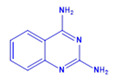	Quinazoline derivative	160.18	150 µM	n.d.	75.9%
DCR 135	C_8_H_7_FN_4_	5-fluoro-quinazoline-2,4-diamine	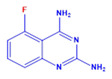	Quinazoline derivative	178.17	250 µM	n.d.	61%
DCR 136	C_8_H_7_FN_4_	7-fluoro-quinazoline-2,4-diamine	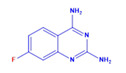	Quinazoline derivative	178.17	250 µM	n.d.	54%
DCR 137	C_8_H_7_FN_4_	6-fluoro-quinazoline-2,4-diamine	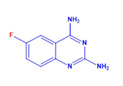	Quinazoline derivative	178.17	300 µM	CC_50_ > 1100 µM, EC_50_ 37 µM, SI 31	90.3%
DCR 138	C_8_H_6_F_2_N_4_	6,7-difluoro-quinazoline-2,4-diamine	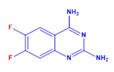	Quinazoline derivative	196.16	250 µM	n.d.	51%
DCR 139	C_9_H_10_N_4_O	5-methoxy-quinazoline-2,4-diamine	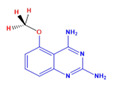	Quinazoline derivative	186.25	250 µM	n.d.	57%

* n.d., not determined. ** Cell viability at 72 hpi as assessed by MTT assay. For cell viability assays, the metabolic activity of the cell control + DMSO (with no drug) was set at 100% as indicated in Figure 1. MNTD and CC_50_ of a compound in Vero cells and EC_50_ of a compound against CHIKV replication in Vero cells were calculated from a dose–response curve of the compound in Figure 2 and Figure 3B, respectively.

## Data Availability

All data including Appendix A was provided.

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
