# Peer review of "Identification of 2,4-Diaminoquinazoline Derivative as a Potential Small-Molecule Inhibitor against Chikungunya and Ross River Viruses"

_viruses, 2023, doi:10.3390/v15112194_

Round 1

Reviewer 1 Report

Comments and Suggestions for Authors

The manuscript authored by Saha and colleagues presents findings that have culminated in the identification of a promising quinazoline derivative, denoted as DCR 137, which exhibited good potency in inhibiting CHIKV replication during screening assays. Furthermore, investigations into DCR 137's antiviral activity against the Ross River virus (an alphavirus) were conducted. In summation, the potential antiviral candidate, DCR 137, merits consideration for subsequent optimization efforts, potentially positioning it as a pan-alphavirus inhibitor.

The significance of investigations directed towards specific antivirals for Chikungunya and Ross River viruses cannot be underrated. Presently, there are no clinically sanctioned antiviral interventions for these viruses, and the therapeutic repertoire remains constrained. Consequently, the advent of efficacious antivirals stands to mark a substantial progression in the management and alleviation of infections instigated by Chikungunya and Ross River viruses. Hence, endeavors directed at formulating targeted antiviral agents for these viruses delineate a pivotal avenue of inquiry in the quest for adept strategies to preclude these burgeoning viral afflictions. 

From my perspective, this study holds relevance within the field, notwithstanding the preliminary nature of the data and the exclusive focus on in vitro analyses. 

I have major points that I see interesting to add to this manuscript:

1. Title: Please considerer writing “Chikungunya and Ross River viruses” instead of “Chikungunya virus and Ross River virus”.

2. Introduction/Discussion: The problematization of Chikungunya disease as an emerging public health concern appears to have been insufficiently investigated. Furthermore, there was an absence of information regarding the Ross River virus.

3. Results: Please, it is imperative to incorporate a flowchart alongside the graphs, elucidating the schematic representation of the experimental steps involved in the antiviral trials (maybe add this in the supplementary file). Existing scientific literature provides various flowchart models for in vitro antiviral assays, encompassing stages such as pre-treatment, co-treatment, and post-treatment.

4. Results: My primary concern lies in the notably high concentration of DCR 137 (i.e., 300 uM) exhibiting antiviral effects at an early stage of testing. Based on my observations in the scientific literature, compounds with recognized antiviral potential have demonstrated efficacy at concentrations well below 300 uM. Could the authors provide additional evidence to establish that they have indeed identified a promising antiviral candidate in this regard?

Author Response

1. Title: Please considerer writing “Chikungunya and Ross River viruses” instead of “Chikungunya virus and Ross River virus”.

[Response]

We really appreciate your valuable advice. As you suggested, we revised the title. Hope our justification and clarification are satisfying to you. 

[Action]

“Identification of 2, 4-diaminoquinazoline derivative as a potential broad-spectrum small molecule inhibitor against Chikungunya and Ross River viruses”

2. Introduction/Discussion: The problematization of Chikungunya disease as an emerging public health concern appears to have been insufficiently investigated. Furthermore, there was an absence of information regarding the Ross River virus.

[Response]

We really appreciate your valuable advice. As you suggested, we added more information regarding Chikungunya and Ross River viruses’ emerging public health concerns in the discussion section. Hope our justification and clarification are satisfying to you.

[Action]

Arthritogenic alphaviruses such as Chikungunya virus (CHIKV) and Ross River virus (RRV) are responsible for large-scale epidemics. These alphaviruses are emerging pathogens that have been progressively expanding their global distribution. CHIKV has recently emerged to cause millions of human infections worldwide. The pandemic potential of CHIKV has long been recognized; in 2018, CHIKV was added to the WHO shortlist for priority research and development, which notably also included pandemic coronaviruses (31). RRV has recently been suggested to be a potential emerging infectious disease worldwide. RRV infection remains the most common human arboviral disease in Australia, with a yearly estimated economic cost of $4.3 billion (32).

References:

31. Bartholomeeusen, K.; Daniel, M.; LaBeaud, D.A.; Gasque, P.; Peeling, R.W.; Stephenson, K.E.; Ng, L.F.; Ariën, K.K. Chikungunya fever. Nat. Rev. Dis. Primers. 2023, 9(1), 17.

32. Yuen, K.Y.; Bielefeldt-Ohmann, H. Ross River virus infection: A cross-disciplinary review with a veterinary perspective. Pathogens. 2021, 10(3), p.357.

3. Results: Please, it is imperative to incorporate a flowchart alongside the graphs, elucidating the schematic representation of the experimental steps involved in the antiviral trials (maybe add this in the supplementary file). Existing scientific literature provides various flowchart models for in vitro antiviral assays, encompassing stages such as pre-treatment, co-treatment, and post-treatment.

[Response]

We really appreciate your valuable advice. As you suggested, we added a flowchart alongside the graphs. Hope our justification and clarification are satisfying to you.

[Action]

Figure 3 (A).

4. Results: My primary concern lies in the notably high concentration of DCR 137 (i.e., 300 uM) exhibiting antiviral effects at an early stage of testing. Based on my observations in the scientific literature, compounds with recognized antiviral potential have demonstrated efficacy at concentrations well below 300 uM. Could the authors provide additional evidence to establish that they have indeed identified a promising antiviral candidate in this regard?

[Response and action]

We totally agree with your concern. As you mentioned, most of the antiviral studies are performed at low inhibitor concentrations preferably below 300 uM. However, there are few exceptional studies where the antiviral potential of a drug has been demonstrated at high concentrations without any cytotoxicity. For example, Aggarwal M. et al. showed efficacy of piperazine against CHIKV at very high concentrations 3-6 mM.  We claim DCR137 as a lead molecule, which may require further structural and biological optimization studies to improve its antiviral potency. Hope our justification and clarification are satisfying to you.

Reference:

Aggarwal, M., Kaur, R., Saha, A., Mudgal, R., Yadav, R., Dash, P.K., Parida, M., Kumar, P. and Tomar, S., 2017. Evaluation of antiviral activity of piperazine against Chikungunya virus targeting hydrophobic pocket of alphavirus capsid protein. Antiviral Research, 146, pp.102-111.

Reviewer 2 Report

Comments and Suggestions for Authors

The authors describe a series of experiments to identify and test a candidate small molecule inhibitor of CHIKV infection with potential broad spectrum anti-alphavirus activity.

The results are extensive and compelling for the lead compound DCR 137 which was applicable to both CHIKV and the closely related RRV. However, extensive editing is required to improve understanding of the methods employeed and the results obtained. Specific comments include:

"Dual acting" in the title is misleading since the mode of action for the compound has not been identified. If it's intended to describe potential broad spectrum antiviral activity the the statement is redundant since CHIKV and RRV are listed in the title.

The methods sections lack sufficient detail. Specifically methods for calculating MNTD, CC50, IC50, SI, percent inhibition, etc. are not detailed. The mode of action studies are incompletey described such that the fate of the inoculum and the treatment is unclear. For pretreatment, was the drug removed, washed away and not replenshed for virus infection? Was the virus inoculum removed following infection? If so, after how long and were the cells washed to remove unattached virus? For simultaneous infection, was the drug removed and not replenished along with the inoculum? For the post infection treatment was the inoculum removed and replaced with medium containing the drug or was the drug simply added to the culture to achieve the desired concentration? Brief descriptions of all methods should be included in the document rather then sighting your previous work.

In the results, MNTD data needs to be described in more detail. It's stated that this analysis was conducted prior to running the antiviral assays but the data is not presented nor was the impact on continued antiviral assessment.

Example of x compounds in the DSS resository, x had MNTDs ranging from x to x. Thus compounds with MNTDs of x and less were excluded from antiviral assessment.

Why is the mass spectrum and purity information important? Shouldn't this have been done beforehand and compounds excluded if they didn't meet these specs?

Why were compounds such as DCR1, 3 and 134 which approached 80% cell viability against an MOI of 1 excluded from further analysis? 

Cell viability assays as presented do not reflect antiviral activity, rather the ability to protect the cells from virus induced CPE. Additionally you might consider presenting this data as % inhibition of of virus induced CPE.

Similarly, fluorescence intensity assays represent inhibition of CHIKV E2 protein expression rather then viral replication specifically. The only direct measurement of impact on virus production is the plaque assays.

Under 3.3, you do not present the results of treatment on RRV induced CPE (Line 75) and you do not discuss the results at 150 uM of compound. It would appear from the fluorescence data that protein expression is further reduced at this concentration when compared to CHIKV and here your looking at polyclonal serum that may represent more then just E2.

For the discussion, conclusions can't be drawn regarding MOA simply because the methods weren't adequately described but the manuscript would also benefit from a discussion of how pre, simultaneous, and post treatments would suggest MOA. The authors suggests the activity is at virus maturation (budding/release) but how does this design rule out inhibition of viral RNA replication? The conclusions for MOA need to be made consistent throughout the discussion because it seems to go back and forth on viral replication vs maturation and release. If it is on replication and release then mention of Mxra8 as a conserved receptor between the two viruses isn't relevant.

Comments on the Quality of English Language

English language is sufficient for comprehension but can be improved throughout.

Author Response

The authors describe a series of experiments to identify and test a candidate small molecule inhibitor of CHIKV infection with potential broad spectrum anti-alphavirus activity.

The results are extensive and compelling for the lead compound DCR 137 which was applicable to both CHIKV and the closely related RRV. However, extensive editing is required to improve understanding of the methods employeed and the results obtained. Specific comments include:

"Dual acting" in the title is misleading since the mode of action for the compound has not been identified. If it's intended to describe potential broad spectrum antiviral activity the the statement is redundant since CHIKV and RRV are listed in the title.

[Response]

We really appreciate your valuable advice. As you suggested, we revised the title. Hope our justification and clarification are satisfying to you.

[Action]

“Identification of 2, 4-diaminoquinazoline derivative as a potential small molecule inhibitor against Chikungunya and Ross River viruses”

The methods sections lack sufficient detail. Specifically methods for calculating MNTD, CC50, IC50, SI, percent inhibition, etc. are not detailed. The mode of action studies are incompletey described such that the fate of the inoculum and the treatment is unclear. For pretreatment, was the drug removed, washed away and not replenshed for virus infection? Was the virus inoculum removed following infection? If so, after how long and were the cells washed to remove unattached virus? For simultaneous infection, was the drug removed and not replenished along with the inoculum? For the post infection treatment was the inoculum removed and replaced with medium containing the drug or was the drug simply added to the culture to achieve the desired concentration? Brief descriptions of all methods should be included in the document rather then sighting your previous work.

[Response]

We really appreciate your valuable advice. As you suggested, we revised the methods section in more details. Hope our justification and clarification are satisfying to you.

[Action]

“The MTT reagent (0.5 mg/ml) was added 72 h post-treatment of compounds. Finally, DMSO was added, and the absorbance was measured at 570 nm. All the readings were normalized with the control experiment in which compounds were not added. The maximum nontoxic doses (MNTD) of all the compounds were calculated. The percentage of cell viability was calculated as follows: 100% − (absorbance of treated cells/absorbance of untreated cells) × 100%. The concentration required to reduce 50% cell viability (CC50) was determined.”

“The dose-response assay was designed to determine the range of efficacy for the DCR137 against CHIKV. The half-maximal effective concentration (EC50) for compounds were calculated. A selectivity index (SI) value was calculated, which is CC50/EC50. Compound with SI value ≥10 is generally considered to be active in vitro.”

“For the pre-treatment group, the compound was added to the cells 24 h prior to infection and the compound was removed by washing the cells before infection; for the simultaneous-treatment group, the compound and CHIKV were administered at the same time; for the post-treatment group, the virus was added to the cells, after 1 h incubation for viral attachment, the inoculum was removed. Finally, the medium containing compound was added 2 h following infection of cells with CHIKV in post-treatment. In the simultaneous and post-treatment modes, the drug was maintained in culture until the supernatants and cells were harvested.”

Figure 3 (A). Flowchart is added.

Legend: Figure 3: Mode of inhibition of DCR 137. (A) Schematic illustration of the time-of-addition/mode of inhibition experiment. Vero cells were infected with CHIKV at an MOI of 1, and treated with DCR137 pre (−24 h), during (0–1 h) and post (2 h) infection. 0.1% DMSO was added at the same time as a control. (B) Post-treatment mode showed significant viral inhibition after treatment with 300 µM DCR 137 compared to pre-treatment and simultaneous- treatment. (C) Inhibition of CHIKV in post-treatment with DCR 137 in a dose dependent manner was assessed by MTT assay at 48 hpi to determine the percent viability after treatment. Data represents the mean ± SD of three independent experiments.

In the results, MNTD data needs to be described in more detail. It's stated that this analysis was conducted prior to running the antiviral assays but the data is not presented nor was the impact on continued antiviral assessment.

Example of x compounds in the DSS resository, x had MNTDs ranging from x to x. Thus compounds with MNTDs of x and less were excluded from antiviral assessment.

[Response]

We really appreciate your valuable advice. As you suggested, we added few sentences regarding MNTD data in the result section. MNTD data of each compound in presented in Table S1. Based on MNTD, compound concentration with more than 90% cell viability was chosen as the highest concentration (MNTD90) for antiviral assessment. Hope our justification and clarification are satisfying to you.

[Action]

‘Prior to evaluating the anti-CHIKV properties of the DCR library, compounds were subjected to toxicity studies in order to determine the maximal dose, which could be non-toxic to the cells. The studies were initiated by using two-fold serially diluted compounds to achieve a specific cytotoxic concentration. Compound concentration with more than 90% cell viability were chosen as the highest concentration (MNTD90). The MNTD of each compound obtained in Vero cells are presented in Table S1. A high MNTD value indicates that the compound is less toxic as compared to the others with low MNTD values. The in vitro antiviral assay was initiated using the MNTD of each compound against CHIKV.”

Why is the mass spectrum and purity information important? Shouldn't this have been done beforehand and compounds excluded if they didn't meet these specs?

[Response and action]

Appreciate the comment. Purity assessment is critical in discovery programs and whenever chemistry is linked with therapeutic outcome. Description of the compound’s chemical constitution, structure, and purity is the key. The mass spec and NMR analysis of all the 150 compounds were done beforehand and we believe that discussing the structure and purity result of at least the best working compound i.e., DCR137 is important. Hope our justification and clarification are satisfying to you. 

Why were compounds such as DCR1, 3 and 134 which approached 80% cell viability against an MOI of 1 excluded from further analysis? 

[Response and action]

Thank you for the reviewer's kind comment and valuable advice. As the reviewer pointed out, we excluded compounds such as DCR1, 3 and 134 which approached 80% cell viability and focused only on the best compound i.e., DCR137 for further validation. We agree with your opinion, and it is something we have planned for future work. Further structural and biological optimization studies are required to improve the antiviral potency of DCR1, 3 and 134. Hope our justification and clarification is satisfying to you.

Cell viability assays as presented do not reflect antiviral activity, rather the ability to protect the cells from virus induced CPE. Additionally you might consider presenting this data as % inhibition of of virus induced CPE. Similarly, fluorescence intensity assays represent inhibition of CHIKV E2 protein expression rather then viral replication specifically. The only direct measurement of impact on virus production is the plaque assays.

[Response and action]

Thank you for the reviewer's kind comment and valuable advice. We opted to represent the cell viability data as it is vital for the evaluation of antiviral efficacy in cell culture system. In the result section we have discussed the data as % inhibition: 

“Percent inhibition with respect to virus control was calculated, and the results showed that the viral titer was reduced nearly to 99.99% and 99.97% at 24 hpi and 48 hpi respectively, in cells post-treatment with 300 μM DCR 137 (Fig. 5A).”

“Percent inhibition calculated with respect to virus control showed 99.99% and 99.85% reduction of virus titer by 300 µM DCR137 at 24 hpi and 48 hpi respectively (Fig. 6D).”

Further, immunofluorescence assay was performed to qualitatively validate the reduction of viral load. 

Under 3.3, you do not present the results of treatment on RRV induced CPE (Line 75) and you do not discuss the results at 150 uM of compound. 

[Response]

Thank you for pointing it out. As suggested, we have now added the results for 150 uM of DCR137. RRV induced CPE at 48 hpi are shown in Fig 6A (first column). Hope our justification and clarification are satisfying to you.

[Action]

“Further, indirect immunofluorescence assay results indicated a low expression of viral proteins in the treated group (300 µM and 150 µM DCR 137) compared to the control in a dose dependent manner, as assessed with RRV-FITC polyclonal antibodies (Fig 6B).”

It would appear from the fluorescence data that protein expression is further reduced at this concentration when compared to CHIKV and here your looking at polyclonal serum that may represent more then just E2.

[Response and action]

It might visually appear like protein expression is reduced more in RRV (polyclonal antibody) when compared to CHIKV (E2), but results were more or less same in both the conditions [and there was consistency in plaque assay data (quantitative data) as well]. The minor disparities might be because we used the best representative images from the respective group (qualitative data). Hope our justification and clarification are satisfying to you.

For the discussion, conclusions can't be drawn regarding MOA simply because the methods weren't adequately described but the manuscript would also benefit from a discussion of how pre, simultaneous, and post treatments would suggest MOA. The authors suggests the activity is at virus maturation (budding/release) but how does this design rule out inhibition of viral RNA replication? The conclusions for MOA need to be made consistent throughout the discussion because it seems to go back and forth on viral replication vs maturation and release. If it is on replication and release then mention of Mxra8 as a conserved receptor between the two viruses isn't relevant.

[Response]

Thank you for pointing it out. As you suggested, we revised the MOA methods section. We have also added information about how pre, simultaneous, and post treatments, are related to initial MOA studies in the discussion section. Appreciate the comment. Some of the sentences are deleted to avoid misunderstanding on viral replication vs maturation and release. Further, Mxra8 is only mentioned to show how the common targets between CHIKV and RRV could be a possibility for broad-spectrum antiviral. There could be some unknown common replication or late replication mediators/target in these viruses responsible for broad-spectrum activity. Hope our justification and clarification are satisfying to you.

[Action]

“Then, a time-of-addition assay was performed to identify which step of viral life cycle is blocked by DCR137. Vero cells were infected with CHIKV at an MOI of 1, and the compound was added before and after infection. Pre-treatment with DCR137 for 24 h prior to CHIKV or simultaneous-infection with virus for 1 h showed no inhibitory effects on viral infection, indicating that DCR137 did not inhibit virus attachment, entry or disassembly process. A maximal reduction in viral titers was observed when DCR137 was added at the 2 hpi post-treatment. These results indicate that DCR137 may exert its effect at early stages of post-entry, such as genome translation and replication (post-entry step of CHIKV life cycle). The specific mechanism of this compound by which it affects post-entry step of CHIKV life cycle effectively is not known, but the possibility that DCR 137 might inhibit a host factor crucial for viral replication.”

“Mechanistically, this study suggests that compound DCR137 inhibits the post-entry step of CHIKV life cycle like genome translation and replication.”

Round 2

Reviewer 1 Report

Comments and Suggestions for Authors

After revisions to the manuscript, the study underwent considerable improvement.

Author Response

We appreciate reviewer for positive evaluation of our work. We have carefully checked and revised the manuscript.

Reviewer 2 Report

Comments and Suggestions for Authors

With the exception of the mode of action studies and figure three the authors have addressed all of the reviewers concerns. It remains unclear how the simultaneous experiment was done. The authors indicated that the virus and compounds were added simultaneously then everything was left on, but the figure suggests that the cells were washed after 1 hour. Was the drug added back to the culture supernatant after washing? If so, why would the results be different from the post-treatment study? Either the method/figure needs to be updated or the discrepancy addressed.

Confounding interpretation is the poor English grammer and punctuation used throughout the document. The information is valuable, but difficult to understand as it is written. For example, what is the definition of post-entry stage? A productive viral infection requires 5 stages including: 1. Attachment, 2. Penetration, 3. Replication, 4. Assembly, and 5) release. The results from pretreatment suggests that the physiology of the cell is not impacted in any way to prevent the five stages of a productive infection. The results from simultaneous treatment, if the drug is removed with the virus, would suggest that the drug does not affect attachment and entry. If the drug is not removed then the data for simultaneous treatment doesn't make sence in comparison to post-treatment. Results from the post treatment experiment suggest that in the presence of the drug virus replication/maturation is impacted. 

Comments on the Quality of English Language

English language needs to be improved throughout and will likely aid greatly in comprehension.

Author Response

Responses to Reviewer’s comments and list of modifications

Journal: Viruses

Manuscript Number: viruses-2629895

Title: Identification of 2, 4-diaminoquinazoline derivative as a dual-acting small molecule inhibitor against Chikungunya virus and Ross River virus

Article Type: Original research paper

Authors: Amrita Saha, Badri Narayan Acharya, Manmohan Parida, Nandita Saxena,

Jaya Rajaiya, Paban Kumar Dash*

We really appreciate reviewers remarks and positive evaluation of our work. We have carefully checked their comments and revised the manuscript accordingly. We hope our revisions will now be satisfactory and acceptable for publication in the ‘Viruses’. Please find our responses to the comments below.

With the exception of the mode of action studies and figure three the authors have addressed all of the reviewers concerns. It remains unclear how the simultaneous experiment was done. The authors indicated that the virus and compounds were added simultaneously then everything was left on, but the figure suggests that the cells were washed after 1 hour. Was the drug added back to the culture supernatant after washing? If so, why would the results be different from the post-treatment study? Either the method/figure needs to be updated or the discrepancy addressed.

[Response]

We are grateful for reviewers’ careful evaluation of our work. As the reviewer pointed out, in the simultaneous-treatment experiments the drug was not added back after washing. It was an error in the schematic, which has been now corrected (we made the schematic for concurrent-treatment by mistake in which the drug is added back to the culture supernatant after washing). Hope our justification and clarification are satisfying to you.

[Action]

As you suggested, we modified the Figure 3A.

Confounding interpretation is the poor English grammer and punctuation used throughout the document. The information is valuable, but difficult to understand as it is written. For example, what is the definition of post-entry stage? A productive viral infection requires 5 stages including: 1. Attachment, 2. Penetration, 3. Replication, 4. Assembly, and 5) release. The results from pretreatment suggests that the physiology of the cell is not impacted in any way to prevent the five stages of a productive infection. The results from simultaneous treatment, if the drug is removed with the virus, would suggest that the drug does not affect attachment and entry. If the drug is not removed then the data for simultaneous treatment doesn't make sense in comparison to post-treatment. Results from the post treatment experiment suggest that in the presence of the drug virus replication/maturation is impacted.

[Response and action]

In pursuance of the reviewer' suggestion, the English writing all over the text has been checked and refined with the help of an expert in English.

Viral life cycle step after virus attachment and penetration is called post-entry stage, which includes genome translation/replication, assembly, maturation, and release. Some of the sentences are now modified to avoid misunderstanding and provide more clarity.

Prevention of viral infection is usually identified by pre-treatment with antiviral agents in cells. Our pre-treatment studies suggest that the DCR137 cannot be used for preventing the CHIKV viral infection (but can be focused on treating the viral infection later). The antiviral effect of simultaneous-treatment means that the initial stages of replication steps (blockage of the binding of virus to cells) were inhibited by treatment. The results from simultaneous-treatment, suggest that the DCR137 does not affect the viral attachment and entry. The inhibition of viral replication is evaluated by post-treatment with antiviral agents after viral infection in cells. The post-treatment experiment suggests that in the presence of the DCR137 viral replication/maturation is impacted.